# MoE Jetpack: From Dense Checkpoints to Adaptive Mixture of Experts for Vision Tasks

**Xingkui Zhu**[*] **Yiran Guan**[*] **Dingkang Liang** **Yuchao Chen**
**Yuliang Liu**[†] **Xiang Bai**
Huazhong University of Science and Technology
`{adlith, yiranguan, dkliang, ylliu, xbai}@hust.edu.cn`

## Abstract

The sparsely activated mixture of experts (MoE) model presents an effective alternative to densely activated (dense) models, combining improved accuracy with computational efficiency. However, training MoE models from scratch requires extensive data and computational resources, a challenge that limits their widespread adoption. To address this, we introduce MoE Jetpack, a framework designed to fine-tune the abundant and easily accessible dense checkpoints into MoE models. MoE Jetpack incorporates two key techniques: (1) *checkpoint recycling*, which initializes MoE models with dense checkpoints to accelerate convergence and enhance accuracy, minimizing the need for extensive pre-training; (2) the *hyperspherical adaptive MoE (SpheroMoE) layer*, which optimizes the MoE architecture to enhance fine-tuning performance and efficiency. Experimental results indicate that MoE Jetpack doubles the convergence speed and enhances accuracy by $2.8\%$ on ImageNet-1K. On smaller datasets, it achieves up to 8-fold faster convergence and over $30\%$ accuracy gains, highlighting its efficiency. The code is available at `https://github.com/Adlith/MoE-Jetpack`.

## 1 Introduction

Increasing model scale is a key factor in boosting deep learning performance [1, 2, 3]. However, as models expand in size, their computational demands surge, resulting in considerable slowdowns during both training and inference phases. A promising approach that decouples model size from computational costs is the *sparsely activated mixture of experts* (MoE) [4, 5, 6]. Unlike *densely activated models* (referred to as *dense models* hereafter) [7, 8] that apply all network parameters to each input, MoE dynamically activates distinct parts of the model based on the input tokens. This allows for model scaling without substantially increasing the FLOPs[2], thereby maintaining training and inference speeds during model upscaling. Recent advancements have seen successful implementations of MoE across various domains [11, 12, 13].

Despite their potential, MoE models face significant adoption challenges primarily due to the lack of pre-trained models. Unlike dense models, which benefit from a vast collection of pre-trained resources available through platforms such as Hugging Face [14] and Timm [15], most MoE models must be trained from scratch with randomly initialized weights. This process demands substantial computational power and large datasets, limiting MoE research to a select few teams with the necessary resources. Consequently, our research aims to reduce the training time and data

---

[*]Equal contribution. † Corresponding author.

[2]FLOPs means the floating point operations per second. The vanilla design of MoE does not inherently provide runtime advantages and requires additional parallelization strategies [9, 10] for acceleration. In our implementation, we offer an effective matrix multiplication method for parallelization, detailed in Appendix C.

38th Conference on Neural Information Processing Systems (NeurIPS 2024).

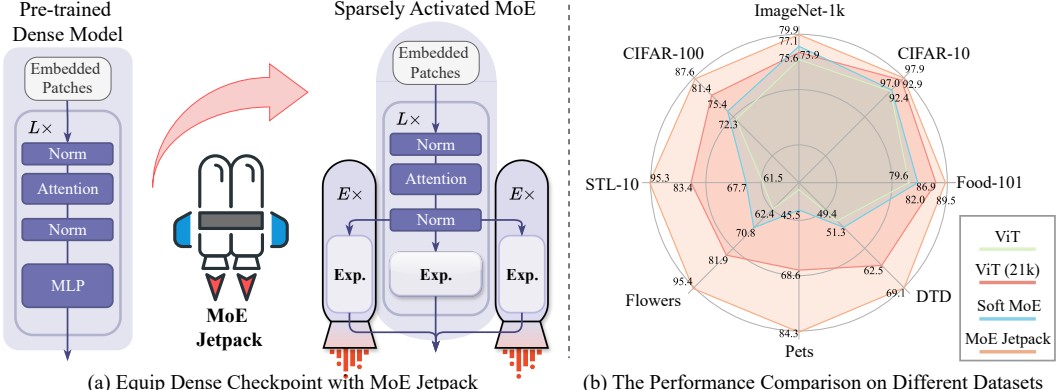

(a) Equip Dense Checkpoint with MoE Jetpack     (b) The Performance Comparison on Different Datasets

Figure 1: (a) MoE Jetpack converts dense checkpoints into initialization weights for MoE models, facilitating faster convergence and improved performance while maintaining equivalent FLOPs. Here, **Exp.** represents individual experts, $E$ denotes the number of experts, and $L$ indicates the total number of layers. (b) Performance comparison among ViT trained from scratch, pre-trained and fine-tuned ViT, Soft MoE [6] trained from scratch, and MoE Jetpack across multiple datasets.

requirements for MoE models by leveraging the pre-trained knowledge from dense checkpoints. *We will specifically investigate whether utilizing dense checkpoints can enhance the accuracy and convergence speed of MoE models during fine-tuning.*

In this paper, we propose MoE Jetpack, a new approach for fine-tuning pre-trained dense checkpoints into MoE models. As illustrated in Fig. 1(a), MoE Jetpack leverages the sunk cost of dense pre-training to boost MoE performance and expedite convergence. It comprises two key techniques. Firstly, **checkpoint recycling** is used to initialize MoE models from dense checkpoints. Unlike sparse upcycling [16], which simply duplicates the Multilayer Perceptron (MLP) to create experts, checkpoint recycling utilizes diverse dense checkpoints and strategic weight selection, providing flexibility and generating higher-quality initialization weights for MoE models. The second technique is the **hyperspherical adaptive MoE (SpheroMoE) layer**, which presents an optimized MoE architecture that facilitates the seamless integration of dense checkpoints and enhances fine-tuning performance. Existing MoE architectures, such as Switch Transformers [4] and Soft MoE [6], are not designed to incorporate pre-trained dense checkpoints, often resulting in optimization inefficiencies and over-specialization during fine-tuning. The SpheroMoE layer mitigates these challenges by normalized token mixing, expert regularization, and adaptive dual-path mechanism, ensuring smoother integration and improved performance.

By equipping dense checkpoints with MoE Jetpack, as illustrated in Fig. 1(b), the fine-tuned MoE models achieve significantly higher accuracy. Comprehensive evaluations across various image classification datasets of different scales further demonstrate the effectiveness of MoE Jetpack. In summary, our contributions are as follows:

- We introduce *checkpoint recycling*, which pioneers the sampling of dense checkpoints to initialize MoE experts, enhancing initialization flexibility, diversifying experts, and eliminating the computational burden of MoE pre-training.

- We develop the *spheroMoE layer*, optimized for fine-tuning dense checkpoints into MoE architectures, alleviating optimization challenges, and preventing the over-specialization of experts.

## 2 Background

In this section, we recap the core concepts needed to understand the MoE Jetpack, specifically focusing on the sparsely activated mixture of experts (MoE) and the routing mechanisms. Existing MoE models are typically derived from dense Vision Transformer (ViT) by replacing certain multilayer perceptron (MLP) layers with MoE layers. Each MoE layer consists of a **Router function**, $\text{Router}(x; \theta_{gate})$, which directs input tokens to a set of "experts", where each expert is parameterized as $\text{MLP}(\cdot; \theta_i)$. Although the experts in MoE models are typically similar, the routing mechanisms vary and are critical to performance. Several routing algorithms have been developed, including top-$k$ [17], BASE and Sinkhorn-BASE layers [18, 19], Hash layers [20], Expert Choice routing [21], and soft routing [6].

The most commonly used mechanism is top-$k$ routing, which reduces computational overhead by selectively activating only the top-$k$ experts most relevant to the input tokens. The routing decision is computed as follows:

$$\text{Router}(x; \theta_{gate}) = \text{top-}k\left(\text{softmax}(\text{MLP}(x; \theta_{gate}))\right), \tag{1}$$

and the output is aggregated by combining the contributions of the selected experts:

$$y = x + \sum_{i \in E} \text{Router}(x; \theta_{gate}) \cdot \text{Expert}(x; \theta_i), \tag{2}$$

where $\theta$ denotes the weights, $E$ is the set of activated experts, and $|E| = K$. However, top-$k$ routing presents challenges such as imbalanced expert utilization, token dropping, and scalability issues.

A more balanced and representative approach, Soft MoE [6], effectively addresses these challenges through implicit soft assignments and serves as our baseline. Instead of assigning tokens to specific experts, Soft MoE computes weighted combinations of all input tokens for each expert. Given the input tokens $\mathbf{X} \in \mathbb{R}^{m \times d}$, where $m$ is the number of tokens and $d$ is their dimensionality, the learnable matrix $\Phi \in \mathbb{R}^{d \times (e \cdot s)}$ project the tokens into $e \times s$ slots, where $e$ is the number of experts and $s$ is the number of slots per expert. The projection is calculated as $\tilde{\mathbf{X}} = \text{softmax}(\mathbf{X}\Phi)^\top \mathbf{X}$. Here, $\tilde{\mathbf{X}} \in \mathbb{R}^{(e \cdot s) \times d}$ represents the transformed inputs that are weighted combinations of $\mathbf{X}$. Each MoE layer includes $e$ expert functions $\{f_i : \mathbb{R}^d \to \mathbb{R}^d\}_{i=1}^e$, with each expert handling $s$ slots. The intermediate outputs $\tilde{\mathbf{Y}} \in \mathbb{R}^{(e \cdot s) \times d}$ are obtained by applying the experts to the slots: $\tilde{\mathbf{Y}}_{i,j} = f_i(\tilde{\mathbf{X}}_{i,j})$ for $i \in \{1, \ldots, e\}$ and $j \in \{1, \ldots, s\}$. Finally, the output tokens $\mathbf{Y} \in \mathbb{R}^{m \times d}$ are reconstructed by combining the expert outputs: $\mathbf{Y} = \text{softmax}(\mathbf{X}\Phi)\tilde{\mathbf{Y}}$.

## 3 MoE Jetpack

In this section, we present the overarching concept of the MoE Jetpack. It is divided into two phases: checkpoint recycling dense checkpoints to initialize MoE models and fine-tuning MoE models using the hyperspherical adaptive MoE (SpheroMoE) layer.

### 3.1 Checkpoint Recycling

Checkpoint recycling is a foundational phase in the MoE Jetpack framework, transforming pre-trained dense checkpoints (**predecessors**) into high-quality initialization weights (**successors**) for MoE models. This approach leverages the rich pre-trained knowledge from predecessors through weight reuse, boosting the performance and convergence speed of successors. The recycling procedure involves sampling a portion of the weights from the predecessors' multilayer perceptrons (MLPs) to construct experts, ensuring expert diversity and adaptability in expert size to meet varied needs.

To define the process of checkpoint recycling (as illustrated in Fig. 2(a)), consider predecessors with $N$ layers, where each layer $L_i$ has a feature dimension of $d$ and a hidden dimension of $4d$. The object is to transform this predecessor into a successor MoE model $S$, which also has $N$ layers, each denoted as $L_i'$, but with a reduced feature dimension $d' \leq d$. Following the Soft MoE architecture [6], the successor comprises two segments: a dense part with $N_1$ layers and an MoE part with $N_2$ layers, where $N_1 = N_2 = \frac{N}{2}$. Formally, the successor model is represented as:

$$S = \left(\{L_i'\}_{i=0}^{\frac{N}{2}-1}, \{L_i'\}_{i=\frac{N}{2}}^{N-1}\right). \tag{3}$$

Inspired by Weight Selection [22], our recycling process ensures consistency in feature dimensions. We explore four primary strategies to guide the recycling of checkpoints:

**Importance-Based Weight Sampling (default)**: We select weights across hidden units and feature dimensions to construct initialization weights for diverse experts. For **feature dimension** selection, maintaining consistency across all layers is essential [22]. We achieve this by calculating the *mean output features* across the $N$ layers and selecting the top-$d'$ dimensions based on these averages:

$$\bar{O} = \frac{1}{N} \sum_{i=1}^{N} O_i, \quad \text{top-}d' = \text{argsort}(\bar{O})[: d'] \tag{4}$$

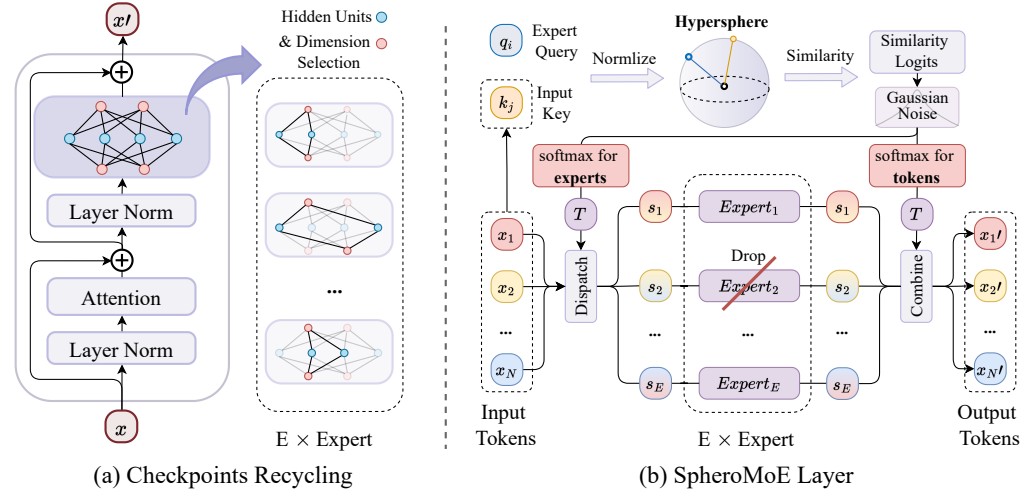

|        (a) Checkpoints Recycling        |        (b) SpheroMoE Layer        |

Figure 2: (a) **Checkpoint Recycling:** Selects hidden units and dimensions from the dense checkpoint MLP, transforming pre-trained weights into diverse experts to initialize MoE models. (b) **SpheroMoE Layer:** Uses cross-attention to dispatch tokens to slots and combine slots back into tokens. Input tokens are projected as keys, and a random query is initialized to compute token-slot similarity in hyperspherical space. Experts then process their assigned slots.

where $O_i$ is the output features of layer $L_i$, and $\bar{O} \in \mathbb{R}^{n \times d}$ is the average output features. For **hidden units**, sampling is performed independently for each layer based on the *magnitude of activations*, as the arrangement of units does not impact the output. Activation values from batches of images are used to form a probability distribution, and units are sampled accordingly:

$$P(h|H) = \frac{A_h}{\sum_{h' \in H} A_{h'}}, \quad h_{\text{successor}} \sim P(h|H), \quad |h_{\text{successor}}| = 4d', \tag{5}$$

where $A_h$ is the activation value of hidden units $h$, and $H$ is the set of all hidden units. This method selects the most important weights for the successor model while promoting diversity among experts through the sampling process.

**Co-Activation Graph Partitioning**: This strategy groups frequently co-activated hidden units into one expert. We construct a co-activation graph by counting the co-activations of hidden units in the testing procedure. Each unit is a vertex in the co-activation graph, and the edges represent their co-activation frequency. Formally, let $G = (V, E)$ be the co-activation graph, where $V$ represents the hidden units and $E$ represents edges with weights indicating co-activation counts. Using the Metis graph partitioning [23], we get several subgraphs:

$$G = \bigcup_{i=1}^{k} G_i, \quad G_i = (V_i, E_i), \quad V_i \cap V_j = \emptyset \text{ for } i \neq j. \tag{6}$$

Experts are formed by the combination of sub-graphs. This method leverages the natural grouping of hidden units, ensuring each expert captures a specific functional subset of the predecessor model.

**Uniform Weight Selection**: Weights are selected uniformly across dimensions and hidden units. For a predecessor with feature dimension $d$ and a successor with dimension $d'$, weights are chosen as:

$$W_{\text{successor}}^{(i)} = W_{\text{predecessor}}^{(k)}, \quad k = \left\lfloor \frac{i \cdot d}{d'} \right\rfloor, \quad i \in \{0, \ldots, d'-1\}. \tag{7}$$

This method ensures an even distribution of the pre-trained weights across the successor MoE.

**Random Weight Sampling**: Weights are randomly selected from the predecessor model. Let $S$ be a random subset of feature dimension indices:

$$S \subseteq 0, \ldots, d-1, \quad |S| = d'. \tag{8}$$

Then, the weights for the successor are chosen as:

$$W_{\text{successor}}^{(i)} = W_{\text{predecessor}}^{(j)}, \quad j \in S, \quad i \in \{0, \ldots, d'-1\}. \tag{9}$$

Through the ablation in Sec. 4.3, Importance-Based Weight Sampling is identified as the default method for recycling dense checkpoints to initialize MoE models.

Notably, the computational overhead introduced by Checkpoint Recycling is virtually negligible. Methods such as Random Sampling and Uniform Selection operate with minimal additional processing, as they directly select experts from the dense checkpoint without further computations. While Graph Partitioning and Importance-Based Sampling involve a preliminary inference step to determine neuron importance or co-activation patterns, the time required is minimal. For instance, performing inference on a subset of $30,000$ images with an RTX 4090 takes less than 5 minutes. This efficient process ensures that Checkpoint Recycling significantly enhances model initialization while introducing almost no additional overhead.

## 3.2 SpheroMoE Layer

Following the initialization of MoE weights through Checkpoint Recycling, the next step is fine-tuning on downstream datasets. To enhance performance and stability, we designed the hyperspherical adaptive MoE (SpheroMoE) layer (Fig. 2(b)), introducing three key improvements: SpheroMoE Routing to alleviate optimization challenges, Expert Regularization to prevent over-specialization, and Adaptive Dual-path MoE (Fig. 3) for better performance and efficiency. Additionally, the pseudo-code detailing these features' implementation can be found in Appendix C.

**SpheroMoE Routing**: As shown in Fig. 2(b), the proposed hyperspherical MoE (SpheroMoE) routing mechanism utilizes cross-attention [24] to distribute inputs across experts. Each expert receives an input slot, a weighted average of all input tokens. To maintain consistency between dense checkpoints and MoE layers $M = \{L'_i\}_{i=N/2}^{N-1}$, input tokens $\mathbf{X} \in \mathbb{R}^{b \times n \times d}$ (where $b$ represents the batch size, $n$ represents the token length, and $d$ represents the input dimension) are layer normalized inherited from dense checkpoints, resulting in $\mathbf{X}_{\text{norm}}$. Queries $\mathbf{Q} \in \mathbb{R}^{b \times (e \times s) \times d}$ are randomly initialized and similarly normalized to align with $\mathbf{X}_{\text{norm}}$, producing $\mathbf{Q}_{\text{norm}}$. The layer normalization process ensures the consistency of distributions between the MoE model, input queries, and the pre-trained dense model. The normalized $\mathbf{X}_{\text{norm}}$ are projected to form keys $\mathbf{K} \in \mathbb{R}^{b \times n \times d}$ for the cross-attention mechanism.

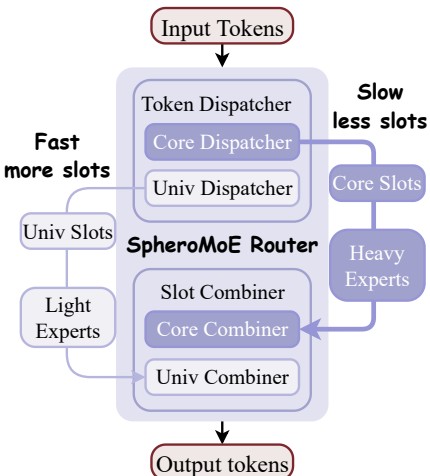

Figure 3: The Adaptive Dual-path MoE structure enhances the SpheroMoE Router by adapting it into a dual-branch system, designed to optimize computational efficiency and model performance. This configuration directs high-impact tokens to a core path with fewer but larger experts while routing less critical tokens to a universal path equipped with a greater number of smaller experts.

To reduce numerical instability, $\mathbf{Q}_{\text{norm}}$ and $\mathbf{K}$ are projected onto a hyperspherical space using L2 normalization, ensuring that the resulting dot products reflect cosine similarities rather than unbounded magnitudes. This confines values within a stable range, preventing softmax saturation and enabling more balanced attention distributions, which improves model generalization. The similarity between $\mathbf{Q}_{\text{norm}}$ and $\mathbf{K}$ is computed, yielding similarity logits $\mathbf{S} \in \mathbb{R}^{b \times (e \times s) \times n}$: $\mathbf{S} = \mathbf{Q}_{\text{norm}} \mathbf{K}^T$. Input slots $\tilde{\mathbf{X}} \in \mathbb{R}^{b \times (e \times s) \times d}$ for experts are formed by a softmax operation along the $n$ dimension of $\mathbf{S}$:

$$\tilde{\mathbf{X}} = \frac{\exp(\mathbf{S}_{ijk})}{\sum_{k'=1}^{n} \exp(\mathbf{S}_{ijk'})} \mathbf{X}_{\text{norm}}. \tag{10}$$

Each expert processes its corresponding input slots $\tilde{\mathbf{X}}_i$ independently, generating outputs $\tilde{\mathbf{Y}}_i$. These outputs are then weighted by $\mathbf{S}$ (after applying a softmax operation along the $(e \times s)$ dimension) to aggregate the experts' contributions, producing the final output $\mathbf{Y} \in \mathbb{R}^{b \times n \times d}$ of the MoE layer:

$$\mathbf{Y} = \frac{\exp(\mathbf{S}_{ijk})}{\sum_{j'=1}^{e \times s} \exp(\mathbf{S}_{ij'k})} \tilde{\mathbf{Y}}. \tag{11}$$

In summary, SpheroMoE routing leverages layer normalization, hyperspherical projection, and cross-attention to effectively distribute inputs across experts, ensuring numerical stability and consistency with pre-trained dense models for improved optimization.

**Expert Regularization**: To improve generalization, SpheroMoE regularizes routing and expert behavior, preventing experts from over-specializing on specific inputs or outputs from depending excessively on certain experts. For the former, we introduce learnable softmax temperatures $T_{dispatch}$ and $T_{combine}$ to precisely control token dispatch and slot combination, enabling smooth transitions between broad and focused attention. Since token dispatch distributes tokens across slots, while slot combination aggregates slot outputs back into tokens, these dual temperatures provide flexible control to each process. Both temperatures are initialized high to promote a broad distribution of attention, preventing early specialization. As the training process, these temperatures adaptively decrease, allowing experts to focus more precisely on relevant features and specialize where advantageous. Additionally, we added a certain level of normal noise to the similarity logits $\mathbf{S}$, which improves generalization. For the latter, we utilized stochastic expert dropout, where each expert $i$ is randomly deactivated with a probability $p$. It ensures that no single expert becomes a crutch for the entire output, promoting a more balanced utilization of all experts. These techniques form an expert regularization strategy that maintains expert versatility and mitigates over-fitting, ensuring the MoE model performs robustly on downstream datasets.

**Adaptive Dual-path MoE**: To mitigate computational redundancy for less critical tokens and concentrate resources on essential ones, SpheroMoE Routing directs input tokens into core and universal slots based on importance. Building on this, the Adaptive Dual-Path structure is designed to assign each slot type to a distinct pathway for optimized processing, as illustrated in Fig. 3. The core pathway consists of a limited number of core experts with larger parameter counts, specifically configured for processing high-importance tokens. In contrast, the universal pathway comprises a larger set of small experts, each with one-fourth of the parameters of core experts, optimized for handling less critical slots. This dual-path configuration improves resource allocation by focusing computation on key tokens, thereby preserving model accuracy and enhancing processing efficiency.

## 4 Experiments

### 4.1 Experimental Setups

**Models.** We validate our approach using the Vision Transformer (ViT)[7] and ConvNeXt[8] models. Specifically, we initialize the weight of V-JetMoE-T and C-JetMoE-F by transforming the dense checkpoints of ViT-S and ConvNeXt-T (pre-trained on ImageNet-21K and sourced from timm) through checkpoint recycling. As detailed in Sec. 3.1, V-JetMoE-T retains the dense layer structure of ViT-T in its first half, while the latter half is equipped with SpheroMoE layers. Each SpheroMoE layer consists of $N/2$ core experts and $N$ universal experts, where $N$ is the number of input tokens. Further details are in Appendix A.

**Datasets.** We evaluate MoE Jetpack on 8 image classification datasets, including ImageNet-1K [25], CIFAR-10, CIFAR-100 [26], Flowers [27], Pets [28], STL-10 [29], Food-101 [30], and DTD [31]. These datasets encompass a diverse range of classification challenges, including object classification, fine-grained species recognition, and texture classification.

**Baseline Implementation.** We follow the implementation details outlined by Xu et al. [22] for comparisons of the dense models. For the MoE models, we employ Soft MoE [6] as the baseline and have replicated it across all datasets. Our MoE Jetpack and Soft MoE utilize the same training strategies as the dense models to ensure comparison fairness. All implementations were executed using the MMPretrain framework [32] on RTX4090. More information can be found in Appendix B.

### 4.2 Main Results

Tab. 1 compares the performance of the MoE Jetpack with Dense ViT models and Soft MoE models on various image datasets using ViT-T (a) and ConvNeXt-F (b) architectures. All models maintain approximately the same number of FLOPs. The columns in Tab. 1 are defined as follows:

- **Dense**: Refers to dense models trained from scratch on each specific dataset.

Table 1: Performance comparison on visual recognition tasks with ViT-T and ConvNeXt-F.

| Dataset (↓) | Dense | Dense (21k) | Soft MoE [6] | MoE Jetpack | Dense | Dense (21k) | Soft MoE [6] | MoE Jetpack |
|---|---|---|---|---|---|---|---|---|
| ImgNet-1k | 73.9 | 75.6 | 77.1 | 79.9 (+2.8) | 76.1 | 76.4 | 79.1 | 80.5 (+1.4) |
| Food-101 | 79.6 | 86.9 | 82.0 | 89.5 (+7.5) | 86.9 | 89.0 | 88.7 | 90.7 (+2.0) |
| CIFAR-10 | 92.4 | 97.0 | 92.9 | 97.9 (+5.0) | 96.6 | 97.4 | 97.3 | 98.2 (+0.9) |
| CIFAR-100 | 72.3 | 81.4 | 75.9 | 88.4 (+12.5) | 81.4 | 84.4 | 82.8 | 88.5 (+5.7) |
| STL-10 | 61.5 | 83.4 | 67.7 | 95.3 (+27.6) | 81.4 | 92.3 | 79.4 | 98.7 (+19.3) |
| Flowers | 62.4 | 81.9 | 70.8 | 95.4 (+24.6) | 80.3 | 94.5 | 83.3 | 98.6 (+15.3) |
| Pets | 25.0 | 68.6 | 45.5 | 84.3 (+38.8) | 72.9 | 87.3 | 77.4 | 94.9 (+17.5) |
| DTD | 49.4 | 62.5 | 51.3 | 69.1 (+17.8) | 63.7 | 68.8 | 64.7 | 79.5 (+14.8) |

(a) ViT-T                                                    (b) ConvNeXt-F

- **Dense (21k)**: Represents dense models initialized with ImageNet-21K pre-trained weights, followed by fine-tuning on the target datasets.
- **Soft MoE**: Reports the results of Soft MoE models trained from scratch on each dataset.
- **MoE Jetpack**: Shows the performance of SpheroMoE models, initialized using checkpoint recycling with pre-trained dense checkpoints from ImageNet-21K, followed by fine-tuning on downstream datasets.

The MoE Jetpack, benefiting from the pre-trained knowledge embedded in dense checkpoints, consistently surpasses the performance of both Soft MoE models trained from scratch and dense models with ImageNet-21K initialization. These results underscore the effectiveness of MoE Jetpack.

### 4.3 Ablations

We perform ablation studies to assess the impact of various components and hyper-parameters within the MoE Jetpack. By default, we use a ViT-T model with the SpheroMoE layer integrated from layers 7 to 12, comprising 98 core experts and 196 universal experts (detailed in Appendix A). The Checkpoint Recycling method transforms dense checkpoints of ViT-S and ViT-T, pre-trained on ImageNet-21k, into initial weights for our V-JetMoE-T model.

**Effect of MoE Jetpack Components.** We conducted the ablation of two key components of the MoE Jetpack on three datasets. As shown in Tab. 2, integrating Checkpoint Recycling with the Soft MoE baseline significantly improves performance across all datasets, with a mean accuracy increment of 10.2%. The SpheroMoE layer further enhances performance, achieving a mean accuracy of 87.9%. These results demonstrate the efficacy of both components, especially when used together, highlighting their synergistic effect in boosting performance.

**Checkpoint Recycling vs. Sparse Upcycling.** To compare the four checkpoint recycling strategies mentioned in Sec. 3.1 and the method of using duplicated MLPs to construct experts in Sparse Upcycling [16], we conducted experiments on ImageNet. For fairness, we also employed our SpheroMoE layer in the Sparse Upcycling. The results, summarized in

Table 3: Checkpoint Recycling vs. Sparse Upcycling

| Method | Construction | ImageNet |
|---|---|---|
| Sparse Upcycling [16] | Copy | 79.1 |
| Checkpoint Recycling | Random Sampling | 79.5 |
| | Uniform Selection | 79.6 |
| | Graph Partitioning | 79.8 |
| | Importance-based Sampling | **79.9** |

Tab. 3, show that Importance-Based Sampling achieves the highest performance, demonstrating its

Table 2: Ablation Study on MoE Jetpack Components.

| Soft MoE [6] | Checkpoints Recycling | SpheroMoE | ImageNet | CIFAR-100 | Flowers | Mean Acc. |
|---|---|---|---|---|---|---|
| Baseline ViT-T | | | 73.9 | 72.3 | 62.4 | 69.5 |
| ✓ | | | 77.1 | 75.9 | 70.8 | 74.6 (+5.1) |
| ✓ | ✓ | | 78.4 | 84.7 | 91.2 | 84.8 (+15.3) |
| | ✓ | ✓ | 79.9 | 88.4 | 95.4 | **87.9** (+18.4) |

Table 4: Effectiveness of SpheroMoE with the Dual-Path Structure.

| Model | Experts | ImageNet | FLOPs (G) |
|---|---|---|---|
| Soft MoE | 197 | 78.4 | 1.2 |
| SpheroMoE | core: 197, univ: 0 | 79.6 | 1.2 |
| SpheroMoE | core: 98, univ: 196 | 79.9 | 1.1 |

Table 5: Comparison of Model Variants with Different Configurations

| model | Weight Init. | MoE Layers | Expert Number | Param (M) | FLOPs (G) | CIFAR-100 | ImageNet |
|---|---|---|---|---|---|---|---|
| ViT-T | - | - | - | 6 | 1.1 | 72.3 | 73.9 |
| Soft MoE-T [6] | - | 7:12 | 197 | 354 | 1.2 | 75.9 | 77.1 |
| Soft MoE-S [6] | - | 7:12 | 197 | 1412 | 4.5 | 77.5 | 80.3 |
| ViT-T | ✓ | - | - | 6 | 1.1 | 81.4 | 75.5 |
| V-JetMoE-T | ✓ | 11:12 | core: 98, univ: 196 | 92 | 1.1 | 87.4 | - |
| V-JetMoE-T | ✓ | 9:12 | core: 98, univ: 196 | 179 | 1.1 | 87.8 | - |
| V-JetMoE-T | ✓ | 5:12 | core: 98, univ: 196 | 352 | 1.2 | 86.7 | - |
| V-JetMoE-T | ✓ | 7:12 | core: 32, univ: 64 | 89 | 0.8 | 87.8 | - |
| V-JetMoE-T | ✓ | 7:12 | core: 64, univ: 128 | 175 | 1.0 | 88.0 | - |
| V-JetMoE-T | ✓ | 7:12 | core: 98, univ: 196 | 265 | 1.1 | 88.4 | 79.9 |
| V-JetMoE-S | ✓ | 7:12 | core: 98, univ: 196 | 1058 | 4.3 | **89.9** | **82.4** |

effectiveness in leveraging critical weights to enhance model performance and convergence speed. Additionally, Checkpoint Recycling is highly flexible, allowing the construction of experts of varying sizes to meet different needs, a feature not provided by sparse upcycling.

**Effect of Dual-Path Structure.** The Tab. 4 presents an ablation study on the effectiveness of the dual-path structure in SpheroMoE. When configured with 197 core experts only (no universal experts), SpheroMoE achieves a higher accuracy on ImageNet-1K compared to Soft MoE with the same number of experts. Introducing the dual-path structure with 98 core experts and 196 universal experts (each with one-fourth of the parameters of core experts) further enhances accuracy to 79.9 while reducing the computational cost to 1.1G FLOPs. This result highlights the efficiency of the dual-path structure, which allows SpheroMoE to allocate resources adaptively and achieve better performance without increasing the overall FLOPs.

**Core Experts Ratio.** To evaluate the effectiveness of the Adaptive Dual-path MoE structure introduced in Sec.3.2 and to identify the optimal ratio between core and universal experts, we conducted ablation on the CIFAR-100 dataset. With a fixed total number of experts, we varied the ratio of core experts to find the ideal balance between performance and resource allocation. As shown in Fig. 4, the highest accuracy is achieved when core experts constitute approximately 1/3 of the total experts.

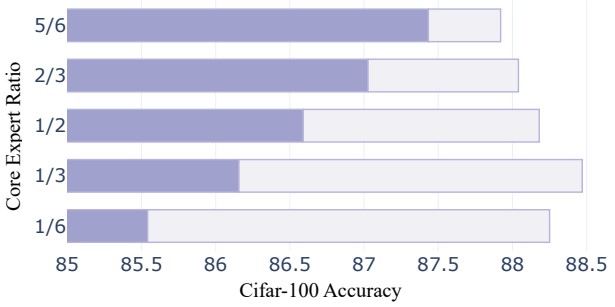

Figure 4: CIFAR-100 accuracy across different ratios of core (dark) to universal (light) experts, highlighting optimal performance at a 1/3 core ratio.

**MoE Jetpack Configurations.** This part evaluates the impact of various MoE Jetpack configurations on model performance, as summarized in Tab. 5. The experiments focus on the placement of SpheroMoE layers, the number of experts per layer, and the base size of converted dense checkpoints. Results indicate that more SpheroMoE layers generally enhance performance, though placing it before layer 7 slightly hurt the performance. Consequently, SpheroMoE layers were incorporated into layers 7−12. Additionally, models with more experts exhibit improved accuracy, highlighting the benefits of increased expert specialization and diversity. Models converted from larger dense checkpoints demonstrate superior performance. These findings suggest that MoE network performance can be improved by increasing the number of MoE layers, incorporating more experts, and utilizing larger base models.

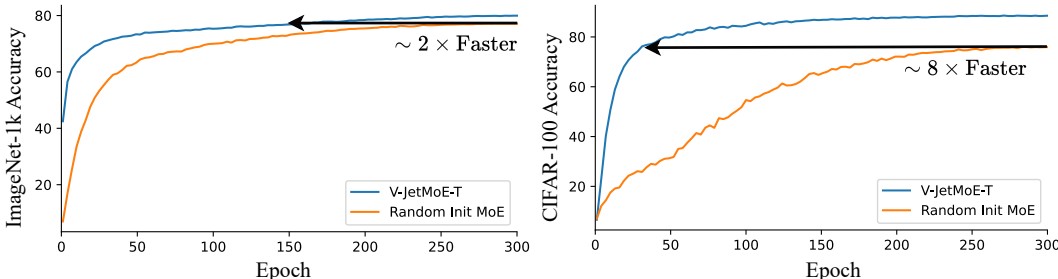

Figure 5: Comparison of convergence speeds using MoE Jetpack versus training from scratch on ImageNet (left) and CIFAR-100 (right). MoE Jetpack achieves target accuracies significantly faster, demonstrating a 2x speed increase on ImageNet and an 8x increase on CIFAR-100.

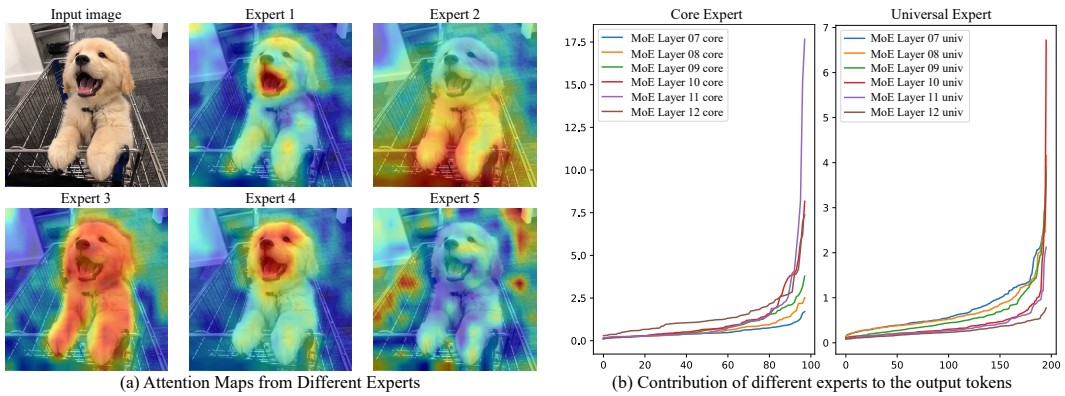

Figure 6: (a) The attention maps generated by five experts in response to the input image, highlighting the experts' specialization. (b) These line charts show varying contributions of core and universal experts, with core experts' influence peaking in later layers, emphasizing their detailed feature refinement, contrasted with the consistent input of universal experts.

## 4.4 Analysis

We analyze the impact of MoE Jetpack on convergence speed for MoE models fine-tuned on ImageNet-1K and CIFAR-100 dataset. Additionally, we offer insights into expert attention patterns and the contribution of each expert to the output tokens.

**Accelerating MoE Convergence with MoE Jetpack.** The effect of MoE Jetpack on convergence speed is illustrated in Fig. 5 for ImageNet (left) and CIFAR-100 (right). In both cases, models with MoE Jetpack achieve target accuracy significantly faster. For ImageNet, MoE Jetpack enables the model to reach approximately 77% top-1 accuracy within 150 epochs—twice as fast as training from scratch. This acceleration is even more pronounced on smaller datasets like CIFAR-100, where MoE Jetpack achieves 76% top-1 accuracy by around 40 epochs, an eightfold improvement over the baseline. These results underscore MoE Jetpack's efficiency in accelerating convergence, reducing fine-tuning time and computational demands.

**Intuition of Expert Attention Patterns.** We visualize the attention maps of experts in Fig. 6(a), which illustrates that different experts focus on different parts of the input image. This diversity in attention suggests that each expert specializes in capturing unique aspects of the input, enhancing the model's ability to represent features comprehensively. The specialization allows the MoE model to combine multiple perspectives, resulting in a more robust and detailed understanding of the input.

**Contribution of Each Expert to Final Results.** Fig. 6(b) demonstrates the varying contributions of core and universal experts across different layers of the MoE model. Core experts show an increasing influence in the later layers, emphasizing their role in refining specific and highly relevant features. Additionally, the contributions among core experts are markedly uneven, some experts can impact output tokens $17\times$ more than others, reflecting greater specialization and diversity in their focus areas.

In contrast, universal experts maintain a relatively consistent contribution level, indicating a more uniform integration of broader contextual information throughout the network. This hierarchical structure, balancing the specialized refinement by core experts with the generalized understanding provided by universal experts, enhances the model's overall performance and robustness.

# 5 Related Work

**Sparsely activated Mixture of Experts (MoE).** Scaling Laws [33] indicate that increasing model parameters can enhance performance. However, traditional densely activated models (dense models) [7, 8] activate all parameters for every input, resulting in high computational costs as models scale. In contrast, MoE models [12, 34, 35, 36] activate only a subset of parameters for specific input tokens, enabling efficient scaling to trillions of parameters with sublinear increases in computational costs [37, 5, 4]. To optimize input token allocation among experts, various routing mechanisms have been developed. BASELayer [18] formulates token-to-expert allocation as a linear assignment problem, while EC-CF2 [21] propose expert choice routing, soft routing methods like SMEAR [38], and Soft MoE [6] implicit soft assignments involving all tokens. However, few studies explore leveraging dense model checkpoints to accelerate MoE training [16].

**Knowledge transfer with pre-trained models.** Knowledge transfer occurs between **identical** or **distinct** models. Pre-training followed by fine-tuning is well-established for **identical models**, utilizing large datasets through supervised learning (e.g., ImageNet21k [25], JFT-300M [39]) or self-supervised methods (e.g., BERT [40], CLIP [41], MAE [42], DINO [43], EVA [44, 45]). These approaches produce foundation models with broad applicability, and subsequent fine-tuning consistently improves performance. For **distinct models**, knowledge distillation [46] trains a smaller student model to mimic the larger teacher model, enhancing efficiency. Additional strategies include weight pruning [47, 48, 48, 49, 50], which removes redundant parameters, and weight selection [22] initializes a smaller model with a subset of weights from a pre-trained larger model.

*Research on transferring knowledge from dense checkpoints to MoE models is limited.* MoEfication [51] partitions a dense model into MoE components, while Sparse Upcycling [16] replicates a dense model multiple times to form a MoE model. Our MoE Jetpack recycles important weights from larger dense checkpoints to initialize experts of various sizes, combining the flexibility of knowledge transfer across distinct model types with the efficiency of transfer between identical model types.

# 6 Conclusion

In this paper, we introduced MoE Jetpack, a novel framework for fine-tuning pre-trained dense checkpoints into Mixture of Experts model. Our approach leverages checkpoint recycling, which inherits the knowledge of open-source dense checkpoints and the hyperspherical adaptive MoE (SpheroMoE) layer to enhance fine-tuning performance. These innovations contribute to improved convergence speed and model accuracy. The MoE Jetpack significantly improved various visual tasks while maintaining computational efficiency.

The **limitation** of our approach lies in its reliance on the quality of pre-trained dense checkpoints; inadequately trained or poorly generalized dense models may hinder performance improvements. Additionally, while our experiments focused on visual tasks, further research is needed to validate the generalizability of MoE Jetpack across other domains, such as natural language processing and reinforcement learning. Future work may address these limitations, further enhancing the scalability and robustness of the framework, and broadening MoE applicability across diverse tasks.

# Acknowledgements

This work is supported by the National Natural Science Foundation of China (Grant Nos. U2341227 and 62226104).

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

# A  Detailed Model Configurations

In this section, we present the detailed model configurations for the main experiments in Sec. 4 in Tab. 6. We refer to pre-trained dense checkpoints as **predecessors** and the derived MoE models as **successors**. We use ImageNet-21k pre-trained predecessor from timm with our Checkpoint Recycling algorithm to generate initialized weights for the successor.

Table 6: Configurations for Models.

| Configuration | Successors | | Predecessors | |
|---|---|---|---|---|
| Model | V-JetMoE-T | C-JetMoE-F | ViT-S/16 | ConvNext-T |
| FLOPs (G) | 1.1 | 1.1 | 1.1 | 1.1 |
| Initialization | Checkpoint Recycling | Checkpoint Recycling | ImageNet-21k | ImageNet-21k |
| MoE Layers | 7:12 | 10:18 | - | - |
| Core Expert Number | 98 | [98, 24] | - | - |
| Universal Expert Number | 196 | [196, 48] | - | - |

# B  Experiment Settings and Time Costs

In this section of the appendix, we provide a comprehensive description of the training settings used in our experiments. Tab. 7 outlines the standard training configuration utilized across our experiments. Tab. 8 details the dataset-specific training configurations, capturing variations in batch size, warmup epochs, total training epochs, and drop path rates for each dataset employed in our experiments.

Our experiments were conducted on RTX 4090 GPU. Training V-JetMoE-T on the CIFAR-100 dataset (60,000 images) required 2.5 GPU hours while training on the ImageNet-1K dataset (1,281,167 images) required 120 GPU hours. Training C-JetMoE-F on CIFAR-100 also required 2.5 GPU hours and 156 GPU hours on ImageNet-1K. For V-JetMoE-S, training on CIFAR-100 required 8 GPU hours and 200 GPU hours on ImageNet-1K. Compared to the original dense models (ViT-Tiny, ConvNeXt-Femto, ViT-Small), our method achieves nearly equivalent training times.

For all the experiments presented in our paper, we required $3,300$ GPU hours for training. In total, we spent approximately $8,000$ GPU hours for exploration and validation of our work.

Table 7: Our basic recipe for model training.

| Training Setting | Configuration |
|---|---|
| image resolution | $224 \times 224$ |
| optimizer | AdamW[52] |
| base learning rate | $4 \times 10^{-3}$ |
| weight decay | 0.05 |
| optimizer momentum | $\beta_1, \beta_2 = 0.9, 0.999$ |
| batch size | 4096 |
| training epochs | 300 |
| learning rate schedule | cosine decay |
| warmup epochs | 50 |
| warmup schedule | linear |
| randaugment [53] | $(9, 0.5)$ |
| mixup [54] | 0.8 |
| cutmix [55] | 1.0 |
| random erasing [56] | 0.25 |
| label smoothing [57] | 0.1 |
| layer scale [58] | $1 \times 10^{-6}$ |

Table 8: Hyper-parameter setting on ViT-T.

| Setting | Batch Size | Warmup Epochs | Training Epochs | Drop Path Rate |
|---|---|---|---|---|
| C-10 | 512 | 50 | 300 | 0.1 |
| C-100 | 512 | 50 | 300 | 0.1 |
| Pets | 512 | 100 | 600 | 0.1 |
| Flowers | 512 | 100 | 600 | 0.1 |
| STL-10 | 512 | 50 | 300 | 0 |
| Food101 | 512 | 50 | 300 | 0.1 |
| DTD | 512 | 100 | 600 | 0.2 |
| IN1k | 4096 | 50 | 300 | 0 |

# C   Implementation of SpheroMoE Layer

Algorithm 1: Simple implementation of SpheroMoE.

```python
def parallel_expert_forward(x, experts)
    """
    Traditional MoE models use a for loop to process each token through the experts.

    By merging all expert weights into a large matrix, our implementation allows
    for a single matrix multiplication operation for each layer across all tokens
    and experts, replacing multiple individual operations.
    """
    x = einsum(x, experts.weight_1, "b e s d1, e d2 d1 -> b e s d2")
    x = x + rearrange(experts.bias_1, "e d2 -> () e () d2")
    x = experts.act(x)
    x = einsum(x, experts.weight_2, "b e s d2, e d1 d2 -> b e s d1")
    x = x + rearrange(experts.bias_2, "e d1 -> () e () d1")
    return x

def spheromoe_layer(X, Q, T, core_experts, univ_experts):
    """
    Performs the Spheromoe layer operation.

    Parameters:
    X (tensor): tensor with shape (batch, token_num, channel).
    Q (tensor): tensor with shape (expert_num, slots_per_expert, channel).
    T (tensor): the learnable temperature list for the softmax function.
    core_experts, univ_experts (expert): expert weight for MoE layer.

    Returns:
    tensor: Output tensor after applying the Spheromoe layer operations.
    """
    X_norm = layer_norm(X, dim=-1)
    Q_norm = l2_norm(Q)
    K = l2_norm(K_project(X_norm))

    # Compute similarity logits.
    logits = einsum(K, Q_norm, "b n d, e s d -> b n e s")

    # create normal noise
    noise = normal_noise(logits) * self.noise_mult

    # Apply softmax to similarity logits.
    Dispatch = softmax(logits/T[0], dim=1) + noise
    Combine = softmax(logits/T[1], dim=[-1,-2]) + noise

    # Token dispatch.
    X_hat = einsum(Dispatch, X_norm, "b n d, b n e s -> b e s d")
    X_core = X_hat[:, :core_num, :, :]
    X_univ = X_hat[:, core_num:, :, :]

    # Using core experts and universal experts processes each slot.
    Y_hat = stack([
        parallel_expert_forward(X_core, core_experts),
        parallel_expert_forward(X_univ, univ_experts)
    ], dim=1)

    # Expert dropout.
    Y_hat = expert_drop(Y_hat)

    # Token combine.
    Y = einsum(Combine, Y_hat, "b n e s, b e s d -> b n d" )

    return Y
```

## D  Dynamic Allocation and Focus Regions of Experts in MoE Jetpack

In this section, we discuss the dynamic allocation and focus regions of core and universal experts across different layers of MoE Jetpack. We used the same test images as in the main text, visualizing the focus regions of the most important (i.e., those with the highest output contribution) core and universal experts for each MoE layer in Fig. 7. The corresponding contribution values for these experts are listed in Tab. 9.

Our findings are as follows: Initially, in the shallower network layers (MoE Layer 7 and 8), the core experts contribute less than the universal experts, and their focus regions are relatively dispersed. As the network deepens, in MoE Layer 9, the most important core and universal experts show similar contribution values and focus regions. With further depth (MoE Layers 10, 11, and 12), the dominance of the core experts becomes increasingly evident, with significantly higher contribution values than the universal experts. Core experts focus on prominent objects in the images and are inclined to capture global information.

These experts' dynamic allocation and different focus region tendencies are crucial to our method. Different experts have varying capabilities in extracting information at various granularities, and the network facilitates collaboration among these experts to produce the final output. This illustrates the effective utilization of expert diversity in the MoE model.

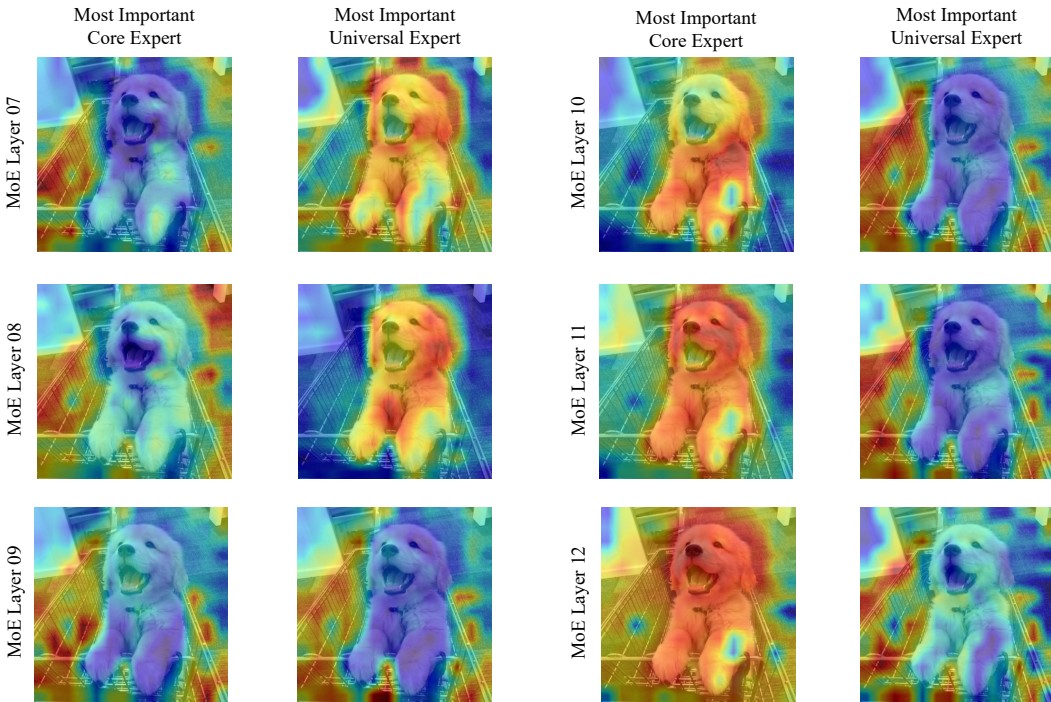

Figure 7: Visualization of the attention map identified by the most important core experts and universal experts across different layers (MoE Layer 07 to MoE Layer 12). The images show the regions deemed most relevant by each type of expert at each layer.

Table 9: Contribution values of core and universal experts across network layers.

| MoE Layer | Core Expert Contribution | Universal Expert Contribution |
|:---:|:---:|:---:|
| 7 | 1.71 | 3.91 |
| 8 | 2.52 | 4.16 |
| 9 | 3.78 | 3.77 |
| 10 | 8.17 | 6.71 |
| 11 | 17.66 | 2.12 |
| 12 | 7.36 | 0.77 |

# E Broader Impacts

The proposed MoE Jetpack framework significantly enhances the accessibility and efficiency of MoE models by utilizing pre-existing dense checkpoints to substantially reduce the computational costs associated with training these models from scratch. This method not only minimizes the environmental footprint by decreasing the reliance on extensive GPU resources but also bridges the resource gap, facilitating wider adoption and fostering innovation across the AI community. Additionally, our commitment to open-sourcing all experimental code promotes greater transparency and collaboration in research. We have carefully considered the potential societal impacts of our method and believe it does not pose any significant ethical or fairness concerns, thereby ensuring its responsible application.

