# OpenReview forum: "MoE Jetpack: From Dense Checkpoints to Adaptive Mixture of Experts for Vision Tasks"
_NeurIPS.cc/2024/Conference — NeurIPS 2024 poster_

### Official Review · Reviewer_hQZu · 2024-06-19

**Soundness:** 3
**Presentation:** 2
**Contribution:** 3
**Rating:** 5
**Confidence:** 5

**Summary:**

The authors propose MoE Jetpack, a method to transform dense checkpoints into a SoftMoE-like network. They introduce checkpoint recycling to utilize dense checkpoints as initial weights for MoE models. Additionally, the authors enhance the SoftMoE with the proposed Expert Regularization and adaptive Dual-path mechanisms. Extensive experiments and ablation studies are conducted to demonstrate the effectiveness of these proposed mechanisms.

**Strengths:**

How to transform a dense checkpoint to an MoE is surely an important problem. The proposed checkpoint recycling method seems promising. Extensive experiments and ablations are given to prove the effectiveness of the proposed method.

**Weaknesses:**

The comparison in the main results appears to be unfair. From my understanding, MoE Jetpack is initialized with a pre-trained model and then fine-tuned on the evaluation dataset. However, the baseline (SoftMoE) is trained from scratch solely on the evaluation dataset. For instance, in the ViT part (Table 1a), MoE Jetpack is initialized with a ViT pre-trained on ImageNet-21k and then fine-tuned on ImageNet-1k, whereas SoftMoE is trained from scratch on ImageNet-1k. This unfairness is even more pronounced on smaller datasets like STL-10 or Pets, as MoE Jetpack already benefits from prior knowledge gained from ImageNet-21k.

**Questions:**

1.	Could you compare MoE Jetpack with a Soft MoE pre-trained on ImageNet21k and fine-tuned on the evaluation dataset?
2.	Could you compare MoE Jetpack with a model pre-trained on ImageNet21k, upcycling to an MoE, and fine-tuned (only router and experts' MLP layer) on the evaluation dataset?

**Limitations:**

Limitation addressed in the paper

---

> ### Author Rebuttal · Authors · 2024-08-04
>
> > **W1-3:** The comparison in the main results appears to be unfair...  MoE Jetpack is initialized with a pre-trained model and then fine-tuned on the evaluation dataset. However, the baseline (SoftMoE) is trained from scratch solely...  you should use a SoftMoE pre-trained on ImageNet-21k and fine-tuned on ImageNet-1k as the baseline. The novelty of the proposed method is limited.
>
> The reviewer's comments seem to reflect a misunderstanding of our methodology and experimental design. We would like to clarify our approach, the **checkpoint recycling and SpheroMoE layer**, and explain the rationale behind our experimental setup.
>
> The background for our method lies in the lack of pre-trained weights for existing MoE models, where pre-training on large datasets like ImageNet-21k requires substantial computational resources. We proposed **checkpoint recycling, a method that leverages pre-trained weights from densely activated models to initialize MoE models, thereby avoiding the significant cost of pre-training.** Inspired by sparse upcycling [1], checkpoint recycling offers greater flexibility and efficiency, allowing for the creation of experts of arbitrary size and number. It is important to note that checkpoint recycling involves selecting and reconfiguring dense checkpoints into experts **without any training or optimization,** resulting in minimal computational overhead. Since checkpoint recycling is a cross-network knowledge transfer method, and specifically aims to circumvent the high computational cost of MoE pre-training, we did not pre-train MoE models on ImageNet-21k.
>
> To demonstrate the effectiveness of checkpoint recycling with SpheroMoE, we used randomly initialized SoftMoE as the baseline in the main results. **To ensure a fair comparison, the influence of initialization was excluded in the ablation study presented in Table 2 of the paper, where the SoftMoE model also utilized the checkpoint recycling technique.** The comparison between sparse upcycling and checkpoint recycling is presented in Table 3 of the paper.
>
> SoftMoE was not designed to utilize dense checkpoints for initialization, and its architecture is **not optimal for the task of checkpoint recycling followed by fine-tuning. Thus, we proposed the SpheroMoE layer, optimized for this task.** SpheroMoE and SoftMoE differ in several significant ways:
>
> 1.  Adaptive Dual-path: Given that checkpoint recycling can produce experts of any size and number, we can control efficiency and performance by using experts of varying sizes. Adaptive Dual-path is introduced to reduce expert parameters while increasing the number of experts, allowing SpheroMoE to achieve better performance with lower FLOPs and fewer parameters.
> 2.  Expert Regularization: This includes a learnable softmax temperature T, expert noise, and stochastic expert dropout, which help mitigate overfitting in MoE models.
> 3.  Extra Norm: To align the distribution between MoE layers and pre-trained models, we use the layer norm from the pre-trained models to constrain the QKV of SpheroMoE. This alignment is crucial for ensuring smooth convergence during fine-tuning, which would not be possible with the original SoftMoE structure. Additionally, to address the impact of the magnitude of randomly initialized queries on similarity calculations and softmax computation, and to introduce regularization to prevent overfitting, we project the queries onto a hyperspherical space using L2 normalization.
>
> The corresponding ablation experiments are presented in the table below. We will revise the manuscript to include this table.
>
> |SoftMoE|Extra Norm|Expert Regularization|Adaptive Dual-path|ImageNet-1k|Cifar100|
> |-|-|-|-|-|-|
> |√| | | |78.4|84.7|
> |√|√| | |79.2|86.9|
> |√|√|√| |79.6|88.2|
> |√|√|√|√|79.9|88.4|
>
> In summary, checkpoint recycling and SpheroMoE work synergistically, leveraging the pre-trained knowledge from dense checkpoints to efficiently fine-tune on downstream tasks without additional pre-training costs. This forms the core motivation behind our method. We hope this explanation provides a clearer understanding of our approach, and we look forward to further discussions.
>
> > **Q1:** Could you compare MoE Jetpack with a Soft MoE pre-trained on ImageNet21k.
>
> **The primary motivation behind our method is to avoid the substantial computational cost of pre-training MoE models on large datasets like ImageNet21k.** Our approach, checkpoint recycling, leverages existing dense checkpoints to initialize MoE models without training, thus bypassing the need for extensive pre-training.
>
> Our work aims to demonstrate that MoE models can achieve competitive performance on downstream tasks without the overhead of pre-training from scratch, which is particularly relevant when computational resources are limited. While comparing MoE Jetpack with a Soft MoE model pre-trained on ImageNet21k would indeed provide additional insights, it falls outside the core objective of illustrating the efficacy of checkpoint recycling.
>
> Nevertheless, we are open to including it in future work, should computational resources permit. This would allow us to explore the full potential of MoE models.
>
> > **Q2:** Could you compare MoE Jetpack with a model pre-trained on ImageNet21k, upcycling to an MoE, and fine-tuned (only router and experts' MLP layer) on the evaluation dataset?
>
> We appreciate the suggestion to explore this fine-tuning mode, which has also been mentioned in ST-MoE [2]. The results of our experiments are as follows:
>
> |MoE Jetpack|ImageNet-1K|
> |-|-|
>  Full fine-tuing|79.9|
> |Only router and experts|78.1|
>
> [1] Komatsuzaki, Aran, et al. "Sparse Upcycling: Training Mixture-of-Experts from Dense Checkpoints." ICLR 2023.
>
> [2] Zoph, Barret, et al. "St-moe: Designing stable and transferable sparse expert models." arXiv:2202.08906.

---

> > ### Comment · Reviewer_hQZu · 2024-08-08
> > **Comment after reading the response**
> >
> > First, I acknowledge the authors' contribution to checkpoint recycling, so I have raised my score accordingly.
> >
> > However, my concerns regarding the fairness of the experiments remain unresolved. I am not the only reviewer who doubts the fairness of the experiments (see R7BZ9 Soundness, Weakness 3).
> >
> > While the authors' proposed method can transfer a dense checkpoint to an MoE without additional computational cost, obtaining the dense checkpoint itself is not free. For instance, in Table 1, Row 8 for the "Pets" dataset, "Soft MoE" only utilizes the Pets data, "Dense(21k)" only utilizes the ImageNet-21k data, and "Dense" only utilizes the Pets data. However, the MoE Jetpack utilizes both ImageNet-21k and Pets data (since the dense checkpoint it initializes from is pre-trained on ImageNet-21k).
> >
> > The MoE Jetpack involves two phases: initialization from a dense checkpoint and fine-tuning (clearly described in lines 83-85). At the very least, the authors should include a baseline of a dense model pre-trained on ImageNet-21k and then fine-tuned on Pets. Reporting "84.3(+38.8)" in Table 1 is misleading. The increase of 38.8 is calculated as 84.3 (MoE Jetpack initialized from a dense model pre-trained on ImageNet-21k, then fine-tuned on Pets) - 45.5 (Soft MoE trained from scratch only on Pets), which exaggerates the improvement. If a stronger dense checkpoint (e.g., pre-trained on CC12M) were used, the gap would be even larger, but such a comparison would not be meaningful.
> >
> > For the same reasons, the comparison of convergence speed is also questionable. It appears the authors are comparing the convergence speed of a model trained from scratch (Random Init MoE) to a pre-trained model (V-JetMoE-T).
> >
> > I look forward to the authors' reply.

---

> ### Author Response · Authors · 2024-08-08
> **Response to Reviewer hQZu**
>
> Thank you for your feedback and for acknowledging the contributions of checkpoint recycling in our work. The statement “If a stronger dense checkpoint (e.g., pre-trained on CC12M) were used, the gap would be even larger” indeed confirms the effectiveness of our checkpoint recycling, indicating that a better pre-trained dense model can further enhance the performance of the MoE model.
>
> However, it appears that your concerns differ from those of Reviewer R7BZ9. Reviewer R7BZ9 suggested **integrating the comparisons in Table 2 into Table 1 to provide a more comprehensive comparison for the readers**. We find this suggestion reasonable and are considering it. We will also discuss further with Reviewer R7BZ9 on how to organize the tables in the paper. On the other hand, you seem to believe that the comparisons in our paper are unfair and that Table 1 is meaningless, as you stated, “such a comparison would not be meaningful.” This is fundamentally different from Reviewer R7BZ9's concern.
>
> **Table 2 in our paper provides a detailed ablation study comparing the results of training SoftMoE from scratch, SoftMoE with checkpoint recycling weights, and SephoreMoE with checkpoint recycling weights.** Table 1, on the other hand, showcases the overall improvements of the entire MoE Jetpack (checkpoint recycling + SephoreMoE). We hope that the ablation study in Table 2 addresses your concerns.
>
> It is worth noting, as mentioned in line 203 of our paper, “Baseline Implementation. We follow the implementation details outlined by Xu et al.” Our experiments were inspired by the work of Xu et al. [1], and Table 1 of our paper adopts a similar experimental setup as theirs. But if you feel that including the comparative results from Table 2 in Table 1 would be beneficial, we would be happy to consider your suggestions.
>
> **Regarding the comparison of convergence speeds in Figure 5, the main improvements are indeed attributed to checkpoint recycling.** Utilizing checkpoint recycling to **transform dense checkpoints into MoE initialization weights significantly enhances both convergence speed and performance, which is a key contribution of our work.** We will further emphasize that checkpoint recycling can significantly accelerate convergence and improve model performance and will also provide a comparison of the convergence speeds of SoftMoE with random initialization and SoftMoE with checkpoint recycling weights to make our paper more comprehensive. Thank you for your valuable suggestion.
>
> Additionally, **I have some questions regarding the reviewer's comments:  "At the very least, the authors should include a baseline of a dense model pre-trained on ImageNet-21k and then fine-tuned on Pets."** Our Table 1, Row 8 already includes this result. Moreover, this result has no connection to MoE. I would appreciate it if the reviewer could further clarify.
>
> We hope our response addresses your concerns and look forward to your reply.
>
> [1] Xu, Zhiqiu, et al. "Initializing models with larger ones." ICLR 2024 (spotlight).

---

> > ### Comment · Reviewer_hQZu · 2024-08-08
> > **Response to Authors**
> >
> > In short, it is unacceptable to compare a pre-trained model (including models initialized from a pre-trained model) directly with a model trained from scratch, especially on small datasets. This is why I maintain that the comparison in Table 1 and Figure 5 is unfair.
> >
> > Consider the results on Pets in Table 1: there are only 7,349 samples in total in the Pets dataset. Training a Soft MoE (ViT-T) on this dataset yields poor accuracy due to insufficient data leading to underfitting. In contrast, MoE Jetpack performs well because its initialization weights are trained on ImageNet-21k. The performance discrepancy between Soft MoE and MoE Jetpack on Pets cannot be attributed solely to checkpoint recycling and SpheroMoE, as a significant portion of this discrepancy stems from the dense checkpoint.
> >
> > To make the comparison more persuasive and appropriate, I suggest using the following baselines in Table 1 instead of Dense, Dense (21k), and Soft MoE trained from scratch:
> >
> > 1. A dense model pre-trained on ImageNet-21k (the same model used for MoE Jetpack initialization) and fine-tuned on Pets.
> > 2. An MoE model initialized from the dense checkpoint using Sparse Upcycling, and fine-tuned on Pets.
> >
> > Additionally, referring to the well-known Sparse Upcycling method[1], all their experiments compare the MoE model with a continuously trained dense model, not a model trained from scratch.
> >
> > [1] Komatsuzaki, Aran, et al. "Sparse Upcycling: Training Mixture-of-Experts from Dense Checkpoints." ICLR 2023.

---

> ### Author Response · Authors · 2024-08-08
> **Response to Reviewer Comments on Sparse Upcycling and Baseline Comparisons**
>
> Thank you for your insightful comments. I would like to address a few points regarding your review.
>
> You mentioned, "referring to the well-known Sparse Upcycling method [1], **all their experiments compare the MoE model with a continuously trained dense model, not a model trained from scratch.**" This statement appears to be problematic. Our experiments in Table 1 were indeed influenced by **Figure 4 in the Sparse Upcycling paper, which is titled: "Figure 4: Pretraining performance achieved by the upcycling method and a MoE model trained from scratch."** I kindly ask you to refer to Figure 4 in the Sparse Upcycling paper for confirmation.
>
> Regarding your suggested baselines:
>
> 1. A dense model pre-trained on ImageNet-21k (the same model used for MoE Jetpack initialization) and fine-tuned on Pets.
> 2. An MoE model initialized from the dense checkpoint using Sparse Upcycling and fine-tuned on Pets.
>
> For the first baseline, **a dense model pre-trained on ImageNet-21k and fine-tuned on Pets, we believe it does not directly relate to our MoE experiments.** Nonetheless, we will conduct this experiment and provide the corresponding results once completed (around 1 day).
>
> The second baseline is essentially an ablation study that compares Sparse Upcycling with our proposed checkpoint recycling method. **This ablation study is already presented in Table 3 of our paper.** To ensure fairness, **both Sparse Upcycling and checkpoint recycling utilize SpheroMoE layers and are compared on the larger ImageNet-1k dataset.** As per your request, we will conduct the experiments on the Pets dataset and include the results in Table 3.
>
> Thank you for your suggestions and thorough review. We will follow up with the additional experiments and provide the updated results accordingly.

---

> ### Comment · Reviewer_7BZ9 · 2024-08-08
>
> Figure 4 in the Sparse Upcycling is indeed titled "Pretraining performance achieved by the upcycling method and a MoE model trained from scratch", but notice that the x axis in that figure only shows the "extra pretraining time".
>
> I think that what reviewer hQZu means is that: Suppose that you train the dense ImageNet-21k model for N steps, and then you use that to initialize the MoE for another M steps, you should compare the performance of the MoE with a dense model trained from scratch on N+M steps.
>
> Table 2 in your paper shows that MoE Jetpack is better than SoftMoE when training both for additional M steps, but it doesn't show if it's simply better to train the original dense model for another M steps.
>
> Of course, this is assuming that all models have the same cost per step, if not one should scale steps proportionally to FLOPs or wallclock time to fairly compare accuracy vs. cost.

---

> ### Author Response · Authors · 2024-08-08
> **Response to Reviewer 7BZ9**
>
> Thank you for your assistance in clarifying this matter. I understand your point regarding the comparison: "You should compare the performance of the MoE with a dense model trained from scratch on N+M steps." This experiment aims to demonstrate that **upgrading a dense checkpoint to an MoE model and training it yields better results than continuously training the dense model**, thereby validating the upcycling process.
>
> The third column in Table 1 (Dense(21k)) represents this process. It shows the performance of a model pre-trained on ImageNet-21K and then fine-tuned on downstream datasets using the same training strategy as MoE Jetpack. Therefore, I am a bit puzzled by the concern since this comparison appears to address the point raised.
>
> Additionally, Figure 4 in the sparse upcycling paper indeed compares sparse upcycling with MoE models trained from scratch. As stated in the paper, "Figure 4 compares sparse upcycling with sparse models trained from scratch. As training from scratch does not reuse the computation cost already sunk into the dense checkpoint..." The x-axis labeled "extra pretraining time" for the MoE curve represents its training time from scratch. If the MoE is pre-trained on the JFT dataset for M steps and then compared with sparse upcycling at N steps on the Extra Pretraining Time, the MoE will obviously achieve much better performance than sparse upcycling.
>
> I hope this explanation clarifies the intended comparisons and the significance of the upcycling process.

---

> > ### Comment · Reviewer_hQZu · 2024-08-08
> > **Response to Authors**
> >
> > 1. The authors claim that the third column in Table 1 (Dense21k) is pre-trained on ImageNet-21k and fine-tuned on downstream datasets. However, I searched for the keyword 'dense' in the paper and did not find any clarification on this point. If I missed something, please point it out.
> >
> > 2. The MoE (green dots) in Figure 4 of the Sparse Upcycling paper is pre-trained on the JFT dataset for M steps (the cost to train the dense checkpoint) and then compared with sparse upcycling at N steps on the Extra Pretraining Time, unless the MoE can achieve 37% accuracy on JFT under random initialization.

---

> ### Author Response · Authors · 2024-08-08
> **Response to Reviewer hQZu comment on Dense21k**
>
> Thank you for your continued communication with us. We apologize for any confusion caused. Our paper mentions, "Tab. 1 compares the performance of the MoE Jetpack with **Dense ViT models (trained from scratch and with pre-trained weights on ImageNet-21k).**" The Dense ViT models with pre-trained weights on ImageNet-21k refer to models that use pre-trained weights from ImageNet-21k and are then fine-tuned on downstream tasks. **We will rewrite this section to: "Tab. 1 compares the performance of the MoE Jetpack with Dense ViT models trained from scratch, and Dense ViT with pre-trained weights on ImageNet-21k then fine-tuning on downstream datasets, represented as Dense (21k) in the table."**
>
> For the second point, **a randomly initialized MoE can achieve 37% accuracy on JFT after a short training period**, as shown in Figure 3 of softMoE [1] and Figure 2 of V-MoE [2]. We are unsure if we have misunderstood your point and look forward to your response.
>
> [1] Puigcerver, Joan, et al. "From sparse to soft mixtures of experts." ICLR 2024.
>
> [2] Riquelme, Carlos, et al. "Scaling vision with sparse mixture of experts." NeurIPS 2021.

---

> > ### Comment · Reviewer_hQZu · 2024-08-08
> > **Response to Authors on Dense21k**
> >
> > Your comment, "The Dense ViT models with pre-trained weights on ImageNet-21k refer to models that use pre-trained weights from ImageNet-21k and are then fine-tuned on downstream tasks," has added to my confusion. Previously, in your **Response to Reviewer Comments on Sparse Upcycling and Baseline Comparisons**, you made a contradictory statement: "For the first baseline, a dense model pre-trained on ImageNet-21k and fine-tuned on Pets, we believe it does not directly relate to our MoE experiments. Nonetheless, we will conduct this experiment and provide the corresponding results once completed (around 1 day)."
> >
> > Please clarify this contradiction, as it raises concerns about the soundness of your experimental results.

---

> ### Author Response · Authors · 2024-08-08
> **Response to Reviewer hQZu comment (Explain the weight selection process)**
>
> To clarify, it is important to distinguish between "using pre-trained weights from ImageNet-21k and then fine-tuning on downstream tasks" and "a dense model pre-trained on ImageNet-21k and fine-tuned on Pets."  There are some differences between the two.
>
> The process of "using pre-trained weights from ImageNet-21k" involves the **ViT-S model, which, after weight selection, leads to the creation of the ViT-T model used for fine-tuning** on downstream tasks. This weight selection [1] step occurs between the pre-training and fine-tuning stages. For detailed information, please refer to the work by Xu et al. As we mentioned in our paper, **"We follow the implementation details outlined by Xu et al. [1] for comparisons of the dense models."**
>
> In contrast, "a dense model pre-trained on ImageNet-21k and fine-tuned on Pets" refers to directly fine-tuning the ViT-T model, which was pre-trained on ImageNet-21k, on the Pets dataset without the intermediate weight selection process from ViT-S to ViT-T.
>
> **Our MoE Jetpack follows the same procedure as Xu et al. [1], incorporating the weight selection process.** This enhances the flexibility of the generated MoE and ensures fair comparisons in Table 1. In contrast, the process of "a dense model pre-trained on ImageNet-21k and fine-tuned on Pets" bypasses the weight selection step from ViT-S to ViT-T and is not closely related to our MoE Jetpack.
>
> We will further modify the description of Table 1 as "Tab. 1 compares the performance of the MoE Jetpack with Dense ViT models trained from scratch, and Dense ViT with pre-trained weights on ImageNet-21k then fine-tuning on downstream datasets with weight selection [1], represented as Dense (21k) in the table."
>
> We hope this clarifies your question. Given that our work builds on numerous previous studies, many details may require referencing prior works for a clearer understanding. We apologize for any confusion caused, thank you for raising the question.
>
> [1] Xu, Zhiqiu, et al. "Initializing models with larger ones." ICLR 2024 (spotlight).

---

> > ### Comment · Reviewer_hQZu · 2024-08-09
> > **Response to Authors**
> >
> > Thank you to the authors and all the reviewers for the thoughtful discussions. After several rounds of back-and-forth, most of my concerns have been addressed, and the experimental results now appear sufficiently sound. I have revised my review and score accordingly.
> >
> > I recommend that the authors include the missing experimental setting details in the final version and reconsider how the results are presented to enhance clarity.

---

> ### Comment · Reviewer_ysa3 · 2024-08-09
>
> Reviewer hQZu has raised several insightful points. However, the discussion has heavily focused on experimental details, potentially diverting attention from the paper's contributions. The motivation of the paper is well-recognized, and the proposed methods are valuable. The main discussion centers on the fairness of the experimental comparisons.
>
> In my opinion, the most crucial aspect is how the experiments demonstrate the effectiveness of the proposed methods. The paper achieves this effectively in several ways:
> - Table 2 shows the SpheroMoE layer more effectively utilizes dense checkpoints than the SoftMoE layer, with improvements from checkpoint recycling.
> - Table 3 indicates that checkpoint recycling is more flexible and effective than sparse upcycling.
> - Table 5 demonstrates that the MoE Jetpack uses dense checkpoints to accelerate model convergence and enhance performance on downstream tasks.
> - Columns 3 and 4 in Table 1 show the combined effects of the SpheroMoE layer and checkpoint recycling, validating their effectiveness.
>
> While the authors are familiar with related work on MoE and have ensured experimental fairness, they need to provide clearer explanations of the experimental settings for each table. Including these in future revisions or supplementary materials will help reduce confusion for meticulous reviewers.
>
> I hope my comments help clarify the discussion.

---

> > ### Author Response · Authors · 2024-08-09
> > **Response to Reviewer ysa3**
> >
> > Thank you for further clarifying the discussion. We appreciate your insights into the effectiveness of our experiments and are pleased that the key aspects of our work are clear.
> >
> > We acknowledge your suggestion to provide clearer explanations of the experimental settings for each table. We will ensure that future revisions or supplementary materials include more detailed descriptions to make our methodologies and results fully transparent and comprehensible to all readers.

---

### Official Review · Reviewer_ysa3 · 2024-07-08

**Soundness:** 4
**Presentation:** 4
**Contribution:** 3
**Rating:** 7
**Confidence:** 5

**Summary:**

This paper introduces MoE Jetpack, a novel method to efficiently fine-tune dense model checkpoints into sparsely activated mixture of experts (MoE) models, reducing the need for extensive data and computational resources. Key techniques include checkpoint recycling and the hyperspherical adaptive MoE (SpheroMoE) layer, which effectively leverage the pre-training costs of existing dense models and optimize the MoE architecture. Experimental results demonstrate significant performance gains and improved convergence speed across various datasets and network architectures.

**Strengths:**

1.	The paper provides a thorough explanation of the methods, including various strategies for checkpoint recycling and detailed architecture of the SpheroMoE layer, offering a clear and replicable approach. The pseudo-code in the appendix is exceptionally clear and well-structured.
2.	The idea of effectively decomposing pre-trained dense models into experts of varying sizes is very interesting. It significantly enhances the flexibility in utilizing the sunk pre-training costs of existing dense models.
3.	The Adaptive Dual-path MoE employs experts of varying sizes within a single MoE model, effectively balancing speed and accuracy. This approach significantly enhances the flexibility and performance of MoE models.
4.	The experiments cover a wide range of datasets and model architectures, ensuring that the results are robust and generalizable.

**Weaknesses:**

1.	The authors need to carefully check the formatting details of the paper:
i) The order of Tables 2 and 3 is incorrect in lines 227 and 262.
ii) There is a missing space before the parentheses in Table 1.
iii) The numerical values within the parentheses in Table 2 need verification for correctness.
2.	The paper misses some related work [2][3][4] and should discuss the proposed method in comparison with these approaches, such as the differences between the proposed Expert Regularization method and the load balancing loss [1] and router Z loss [2]. Are the methods proposed in this paper orthogonal to these existing methods?
3.	The additional computational overhead introduced by checkpoint recycling should be detailed and quantified by the authors.

[1] Fedus W, Zoph B, Shazeer N. Switch transformers: Scaling to trillion parameter models with simple and efficient sparsity[J]. Journal of Machine Learning Research, 2022, 23(120): 1-39.
[2] Zoph B, Bello I, Kumar S, et al. St-moe: Designing stable and transferable sparse expert models[J]. arXiv preprint arXiv:2202.08906, 2022.
[3] Xue F, Zheng Z, Fu Y, et al. OpenMoE: An Early Effort on Open Mixture-of-Experts Language Models[C]//Forty-first International Conference on Machine Learning.
[4] Zhou Y, Du N, Huang Y, et al. Brainformers: Trading simplicity for efficiency[C]//International Conference on Machine Learning. PMLR, 2023: 42531-42542.

**Questions:**

1.	I am curious about the differences that might arise from initializing expert weights from models of different sizes. Is a larger model always better?

---

> ### Author Rebuttal · Authors · 2024-08-05
>
> We sincerely thank the reviewers for their valuable comments and suggestions. We have carefully considered each point and provided our responses below.
>
> > **W1:** The authors need to carefully check the formatting details of the paper...
>
> We acknowledge the formatting issues identified. We will correct the order of Tables 2 and 3, ensure the numerical values within the parentheses in Table 2 are verified, and address the missing space before the parentheses in Table 1. These corrections will be implemented in the revised manuscript.
>
> > **W2:** The paper misses some related work and should discuss the proposed method in comparison with these approaches...
>
> We agree that a more thorough comparison with related work is necessary. We will expand the related work section to include these papers. In ST-MoE, the load balancing loss and router-Z loss are employed. The load balancing loss, however, is not applicable to SpheroMoE due to its use of a cross-attention-like mechanism that dynamically assembles the input for each expert. This is an advantage of the SoftMoE design, which inherently balances the load without needing additional loss functions.
>
> The router-Z loss, on the other hand, operates on the logits of the gating mechanism. It aims to regulate the scale of the logits, encouraging them to remain as small as possible in absolute value, as larger logits can lead to larger gradients within the softmax activation function, potentially causing instability during training. This loss can indeed be applied to SpheroMoE to constrain the logits and enhance numerical stability.
>
> We will conduct further experiments to investigate the impact of incorporating router-Z loss in SpheroMoE and include the findings in the revised manuscript. We appreciate the reviewer's valuable suggestion and will ensure that these aspects are thoroughly explored.
>
> > **W3:** The additional computational overhead introduced by checkpoint recycling should be detailed and quantified by the authors.
>
> We understand the concern regarding the additional computational overhead introduced by checkpoint recycling. Checkpoint recycling involves four methods for weight selection: Random Sampling, Uniform Selection, Graph Partitioning, and Importance-based Sampling. The first two methods, Random Sampling, and Uniform Selection, **do not entail any additional computational processes**, as they directly select experts from the MLP of a larger network without further calculations.
>
> In contrast, Graph Partitioning and Importance-based Sampling require a preliminary step. We first randomly select a small subset of images (30,000) from ImageNet 1k and perform inference using the dense checkpoint on these images. This step is necessary to obtain the intermediate activation values required for determining the graph partitioning and assessing the importance of individual neurons. For instance, when selecting experts for the V-JetMoE-T model, we use ViT-S to infer on the selected images. As this process does not involve optimizing the network, the computational overhead is minimal. Specifically, on an RTX 4090, **this inference process takes less than 5 minutes.**
>
> We will include these details and a quantitative analysis of the overhead in the revised manuscript to provide a clearer understanding of the computational impact of checkpoint recycling. Thank you for pointing this out.
>
> > **Q1:** I am curious about the differences that might arise from initializing expert weights from models of different sizes. Is a larger model always better?
>
> This is a highly insightful question. **While larger models generally contain more extensive pre-trained knowledge, initializing experts from them does not always yield better performance** in a fixed-size MoE network (e.g., V-JetMoE-T). In practice, experts selected from models of similar sizes (e.g., ViT-S) often perform better. This observation aligns with previous findings in weight selection [1, 2] and knowledge distillation [3, 4].
>
> Larger models, though more expressive, may possess a high degree of redundancy in their learned representations. When transferring weights to a smaller model, **the size disparity can lead to a loss of critical structural information,** thereby diminishing the benefits of the pre-trained knowledge. Moreover, **smaller models closer in size to the target MoE network often have more compatible feature representations,** which are easier to adapt and fine-tune within the MoE framework. This compatibility can lead to more efficient utilization of the model's capacity, resulting in better generalization and performance. Therefore, selecting experts from models of similar size helps maintain a balance between the richness of learned features and the adaptability of the network's capacity.
>
> [1] Xu, Zhiqiu, et al. "Initializing models with larger ones." ICLR 2024.
>
> [2] Pan, Zizheng, Jianfei Cai, and Bohan Zhuang. "Stitchable neural networks." CVPR 2023.
>
> [3] Hu, Chengming, et al. "Teacher-student architecture for knowledge distillation: A survey." arXiv preprint (2023).
>
> [4] Pham, Cuong, Tuan Hoang, and Thanh-Toan Do. "Collaborative multi-teacher knowledge distillation for learning low bit-width deep neural networks." WACV 2023.

---

> ### Comment · Reviewer_ysa3 · 2024-08-12
>
> Thank you for your detailed responses, which effectively address my concerns. Consequently, I maintain my positive rating. The expanded discussion on router-Z loss, alongside the comprehensive analysis of the computational overhead, will significantly enhance the manuscript's clarity and depth. I look forward to seeing the revised manuscript.

---

> ### Author Response · Authors · 2024-08-12
>
> Thank you for your positive feedback and constructive comments. we are pleased that our rebuttal addressed your concerns. We appreciate your suggestions and will incorporate them into the revised version.

---

### Official Review · Reviewer_qYPk · 2024-07-12

**Soundness:** 3
**Presentation:** 3
**Contribution:** 3
**Rating:** 5
**Confidence:** 3

**Summary:**

This paper introduces MoE Jetpack, a novel method for fine-tuning pre-trained dense model checkpoints into mixture of experts (MoE) models. It leverages checkpoint recycling to accelerate convergence and enhance accuracy, and incorporates a hyperspherical adaptive MoE (SpheroMoE) layer for optimized integration and performance. Extensive evaluations show that MoE Jetpack not only speeds up the fine-tuning process but also achieves higher accuracy across various datasets while maintaining computational efficiency.

**Strengths:**

1. Tables and figures are informative, and the paper is generally well written.
2. The technical contributions of the paper are substantial.
3. The experiments comprehensively validate the effectiveness of the proposed methods.

**Weaknesses:**

1. The work consumes extensive computational resources to validate a complex algorithm on small-scale datasets. Specifically, all experiments presented required 3300 GPU hours for training and a total of 8000 GPU hours for exploratory purposes. However, the datasets primarily showcased, aside from ImageNet, are of small scale and outdated. Given the current emphasis on large models, these small datasets are not suitable for substantiating the claims of this paper. It is recommended to conduct more extensive experiments on ImageNet or similar large-scale datasets to validate scalability and justify the extensive use of pre-trained checkpoints.

2. From an application perspective, the effectiveness of utilizing checkpoints in practical scenarios is questionable. For instance, the core result of the ViT-based models, where V-JetMoE-T with 264M parameters only achieves 79.9% accuracy, suggests that using pre-trained weights for MoE may not be as parameter-efficient as expected. Instead, using comparable computation models, like those on the ImageNet performance leaderboard, might be more advantageous. This critique underscores the need to demonstrate that utilizing pre-trained weights for MoE can surpass the performance of similarly scaled models trained conventionally.

**Questions:**

Please refer to the concerns raised in the weaknesses section. It is essential to provide more experiments to demonstrate that MoE models using pre-trained weights deliver significantly better performance, and to verify that the proposed method has sufficient scalability.

**Limitations:**

The paper has discussed the limitations in the conclusion, specifically noting that the method is only applicable to visual tasks. There is no need to consider potential negative societal impacts for this work.

---

> ### Author Rebuttal · Authors · 2024-08-03
>
> Thank you for your review and constructive comments on our paper. We appreciate the opportunity to address your concerns and clarify our work.
>
> > **W1:** The work consumes extensive computational resources on small-scale datasets.  It is recommended to conduct more extensive experiments on ImageNet or similar large-scale datasets...
>
> We appreciate the reviewer's suggestion to conduct more comprehensive experiments on large-scale datasets. While we agree that such experiments would further substantiate the efficacy of our approach, our current hardware setup, consisting of 8 RTX 4090 GPUs, imposed significant limitations. The (3300 + 8000) GPU hours expended represent the maximum computational budget we could allocate, **with 70% of this time dedicated to ImageNet-1k.** The primary experiments and most of the ablation studies conducted on ImageNet-1k effectively demonstrate the robustness of our method. **ImageNet-1k has been widely recognized and utilized in recent literature as a standard benchmark for validating new methods [1,2,3,4].** Should more computational resources become available in the future, we plan to further validate our approach on additional and larger datasets.
>
> Additionally, **the experiments conducted on smaller datasets, although not consuming a significant portion of computational resources, play a crucial role in illustrating the adaptability and generalization capabilities of our method.** In practical scenarios, many downstream vision tasks may not have access to extensive datasets like ImageNet-1k. Therefore, evaluating our approach across datasets of varying scales offers valuable insights into its performance under conditions of limited data availability. This not only highlights the versatility of our method but also its potential applicability in diverse real-world scenarios.
>
> [1] Xu, Zhiqiu, et al. "Initializing models with larger ones." ICLR 2024.
>
> [2] Chen, Mengzhao, et al. "Diffrate: Differentiable compression rate for efficient vision transformers." CVPR, 2023.
>
> [3] Pan, Zizheng, Jianfei Cai, and Bohan Zhuang. "Stitchable neural networks." CVPR, 2023.
>
> [4] Trockman, Asher, and J. Zico Kolter. "Mimetic initialization of self-attention layers." ICML, 2023.
>
> > **W2:** From an application perspective, the effectiveness of utilizing checkpoints in practical scenarios is questionable. ... V-JetMoE-T with 264M parameters only achieves 79.9% accuracy...
>
> Thank you for your thoughtful comments on the parameter efficiency of MoE architectures compared to dense models, particularly those on the ImageNet performance leaderboard.
>
> **It is important to note that the primary consideration here is not merely the checkpoint recycling but rather the fundamental differences between MoE and dense architectures.** Moreover, the effectiveness of utilizing checkpoints in practical scenarios has already been demonstrated in the Sparse Upcycling paper [1], with specific results detailed in Appendix C. The scalability of MoE models has been evidenced in numerous prior works [1,2,3]. For instance, the Soft MoE L/16 128E model, with 13.1B parameters, achieved an accuracy of 89.2% on ImageNet. However, conducting experiments of this scale is far beyond our available resources.
>
> MoE models function under a sparse activation paradigm, which is quite distinct from dense models. Specifically, MoE models are designed to activate only a subset of parameters, thus optimizing computational resources and enabling scalability without a linear increase in computational costs. For instance, **the V-JetMoE-T model (265M parameters, 79.9% accuracy, 1.1G FLOPs) shares the same FLOPs as the ViT-T model (6M parameters, 75.6% accuracy, 1.1G FLOPs), their training and inference time are very similar.** This is a key advantage of MoE models: they can improve performance while maintaining comparable FLOPs, making them well-suited for scenarios that require efficient training and inference.
>
> Furthermore, when **given the same training budget, MoE Jetpack can often achieve superior performance compared with vanilla ViT.** For example, V-JetMoE-T reaches 79.9% accuracy with almost the same computational overhead as ViT-T, which only achieves 75.6% accuracy (considering the fine-tuning process). **A dense model with a similar parameter of V-JetMoE-T (256M) would require approximately 54G FLOPs, resulting in a significantly higher computational cost**—around 50 times that of V-JetMoE-T. Therefore, direct comparisons between the parameter counts of dense and MoE models may not be entirely fair.
>
> We hope this explanation clarifies the distinctions and advantages of MoE models. Thank you again for your valuable feedback.
>
> [1] Komatsuzaki, Aran, et al. "Sparse upcycling: Training mixture-of-experts from dense checkpoints."  ICLR 2023.
>
> [2] Puigcerver, Joan, et al. "From sparse to soft mixtures of experts." ICLR 2024.
>
> [3] Riquelme, Carlos, et al. "Scaling vision with sparse mixture of experts." NeurIPS 2021.

---

### Official Review · Reviewer_7BZ9 · 2024-07-28

**Soundness:** 2
**Presentation:** 3
**Contribution:** 3
**Rating:** 6
**Confidence:** 5

**Summary:**

The paper proposes several ideas to convert dense checkpoints into MoEs for vision tasks. In particular, four alternatives are presented to reuse checkpoints of dense models to be use as an initial checkpoint for an MoE model for a later fine-tuning phase. Of the four alternatives presented, Importance-Based Weight Sampling is the one that works the best. Most importantly, all of them are reported to work better than Sparse Upcycling, at least when evaluated on ImageNet (see Table 3).
The second novelty that the paper presents is a variant of the SoftMoE router: the SpheroMoE router. The SpheroMoE has an additional projection on the inputs, and places the L2 normalization in different places, compared to SoftMoE. It also has a temperature parameter that is decayed over time. Similar to the SoftMoE router, the SpheroMoE computes slots as a weighted average of the input tokens (similar to attention).
Last but not least, the paper introduces the Adaptive Dual-Path structure in the MoE blocks. The experts are divided in two sets: a set of light experts that process lots of slots, and a set of heavy experts that process fewer slots.
The paper presents results on variety of common vision tasks, such as ImageNet-1k (a.k.a. ILSVRC2012), CIFAR-10/100,  Food-101, etc. Sometimes the models are trained from scratch on these datasets, and others an initial (dense) checkpoint pre-trained on ImageNet-21k is used.

**Strengths:**

- The different ideas presented to initialize a MoE from a dense checkpoint are new and are shown to work better as a previously proposed approach: Sparse Upcycling. The explanation of the methods is well-detailed in the paper.
- A quite detailed implementation of the SpheroMoE router is given in the appendix, which will certainly widen the reproducibility and potential adoption of the paper.
- Most of the presented ideas have been ablated on different experiments. In particular, the choice of Importance-Based Weight Sampling is justified in Table 3, experimenting on ImageNet, which is a quite rich dataset. The use of the SpheroMoE router is well justified in Table 2, and is showed to be superior to the SoftMoE router on 3 datasets (ImageNet, CIFAR-100, Flowers).
- Overall, it's shown that using MoE Jetpack not only significantly accelerates the training on ImageNet-1k and CIFAR-100, compared to training from scratch, but also the models achieve a higher accuracy which would probably not be achieved from scratch, or at least not in a reasonable amount of time, in the case of CIFAR-100.

**Weaknesses:**

- All experiments are done with quite modest backbones: Tiny (T) and Small (S) vision tranformer backbones. The Sparse Upcycling paper, that the authors use a baseline, observes that the benefits of initializing from a dense checkpoint are much smaller when using larger backbones (especially when evaluating transfer to small downstream tasks), see Figure 3 and 4 from https://arxiv.org/pdf/2212.05055. So,  it's not clear at all wheather the proposed improvements will make a difference at much bigger scales.
- SpheroMoE has far more experts that SoftMoE. It's not clear wheather the benefits come from the improvements or by using finer-grained & more experts, which has been proposed before (for instance, in https://arxiv.org/abs/2401.06066 and https://arxiv.org/abs/2402.07871).
- The results for SoftMoE in Figure 1b and Table 1 are for models trained from scratch. Since a distinction is made for ViT on whether it's trained from scratch or pre-trained on ImageNet-21k, perhaps it would be fairer to present this distinction for SoftMoE as well in these main figure and table. This ablation is presented in Table 2, but only for 3 datasets.
- Figure 2b and Equation 10 (and algorithm 1 in the appendix), do not quite match in one aspect: the L2-normalized X is also used to compute the content of the input slots of each expert, and not the original X as Figure 2b seems to depict.
- SpheroMoE has a few differences with SoftMoE. It would be much better if the individual differences were ablated. For instance, what's the effect of the additional (linear?) projection on X used to compute the combine and dispatch weights? What's the difference between having a decayed temperature vs. having a learned temperature (or scale) as SoftMoE does?

**Questions:**

- In Algorithm 1, the meaning of `K_project` and `inherit_layer_norm` are not clearly explained. Please, do so. In particular, what's the output size of the `K_project`? Is it the same as the embedding size of X?
- In that regard, assuming that  `inherit_layer_norm` is doing some sort of LayerNorm with shared parameters, LayerNorm and L2-normalization are very similar, except for a $\sqrt{n}$ factor and the additional $\alpha$ or scale parameters of LayerNorm (and potentially the biases too). How was the use and positioning of the additional L2-norm decided?

**Limitations:**

No negative societal impact is particular to this work, in my opinion.

---

> ### Author Rebuttal · Authors · 2024-08-01
>
> Thank you for your detailed review of our paper and for highlighting the novelty and effectiveness of our method. We will address each of the concerns you raised in detail.
>
> > **W1:** All experiments are done with quite modest backbones ...
>
> We acknowledge the use of modest backbones due to limited computational resources, as our experiments were conducted on 8 NVIDIA RTX 4090 GPUs. Training a vanilla ViT-S model on ImageNet (1.28M images) took over two days, limiting our ability to explore larger models.
>
> The Sparse Upcycling paper suggests that the benefits of dense checkpoint initialization diminish with larger models, likely due to diminishing marginal returns on performance for smaller tasks. However, our MoE Jetpack improves upon Sparse Upcycling by not only using dense checkpoints but also modifying the MoE architecture, which we expect to yield better results.
>
> Despite the constraints on large-scale experiments, our proposed checkpoint recycling and SpheroMoE layer present promising avenues. We conducted experiments within our means using modest backbones, which can still provide valuable insights and demonstrate the potential of our approach.
>
> > **W2:** SpheroMoE has far more experts...
>
> In our ablation study, we ensured a fair comparison between SoftMoE and SpheroMoE by increasing the number of experts in SoftMoE to 6*197, rather than using the original count from the SoftMoE paper. However, we did not compare the models with an equal number of experts. The corresponding results are provided in the table below and will be included in the revised manuscript. When the number of universal experts (experts with 1/4 parameters) in the dual-branch structure is set to zero, SpheroMoE and SoftMoE have the same number of experts.
>
> |Model| Number of Experts | ImageNet 1k | FLOPs(G) |
> |-|-|-|-|
> |SoftMoE|197|78.4|1.2|
> |SpheroMoE|core: 197, univ: 0 |79.6|1.2|
> |SpheroMoE|core: 98, univ: 196|79.9|1.1|
>
> > **W3:** SoftMoE are trained from scratch in the main result...
>
> We designed the main table to highlight the overall performance improvements from our two approaches: checkpoint recycling and SpheroMoE. Using SoftMoE trained from scratch as the baseline effectively demonstrates the combined effectiveness of checkpoint recycling and SpheroMoE. Therefore, placing the ablation study of SoftMoE and SpheroMoE, both initialized with checkpoint recycling, in Table 2 is more appropriate to illustrate the component effectiveness.  We conducted this ablation on three representative datasets, including ImageNet, and will aim to conduct more extensive experiments on additional datasets.
>
> > **W4:** Figure 2b does not quite match.
>
> Some details were missing in Figure 2. We will incorporate additional details in the subsequent versions.
>
> > **W5 & Q1:** SpheroMoE has a few differences with SoftMoE.  1) The effect of `key_proj`. 2) decayed temperature.
>
> (1) **Additional Projection:**
> In cross-attention, queries (Q), keys (K), and values (V) are generated by applying linear projections. In SpheroMoE, projections are adjusted to align the activation distributions between the SpheroMoE layer and the pre-trained model.
>
> - **V Projection:**
> To maintain alignment with pre-trained dense checkpoints, we use layer normalization from the pre-trained network to process input $X$ to produce $V$. The `inherit_layer_norm` in the pseudocode corresponds to `self.norm2` below.
> ```
>     # vit block forward
>     x = x + self.mlp(self.norm2(x))
> ```
> - **Q and K Projections:** For K, we used a scale-and-shift as `k_proj`: $K = \alpha_k \cdot X + \beta_k$, while Q underwent layernorm, containing scale and shift adjustments. On ImageNet, these projections didn't help. However, on smaller datasets, these projections both accelerated convergence and enhanced final performance.
>
> (2)  **Decayed Temperature:**
> We apologize for the limited explanation provided in the manuscript.   **SoftMoE** uses a single trainable scalar for scaling Q: $\text{scale} \cdot Q$. In **SpheroMoE**, the Decayed Temperature $T$ is represented as a list with two learnable parameters, which adjust the dispatch and combine logits, respectively:
>
> $Dispatch = softmax(logits/T[0] + noise, dim=1)$,
>
> $Combine = softmax(logits/T[1] + noise, dim=[-1,-2])$.
>
> $T$ is initialized to large values (1.5) to ensure a uniform distribution of attention among experts. As training progresses, these coefficients are optimized by backpropagation and gradually decrease. The ablation is in the table below.
>
> |SoftMoE|ImageNet 1k|
> |-|-|
> |Scale|78.4|
> |$T$|78.7|
>
> > **Q2:**  LayerNorm and L2-normalization are very similar. How ... decided?
>
> While LayerNorm and L2-normalization share similarities, they serve distinct purposes. LayerNorm adjusts the mean and variance, stabilizing activations, while L2-normalization scales input to unit norm.
>
> We use layer_norm on the query (Q) to maintain a consistent distribution with the keys (K).
>
> L2-norm is applied for the following reasons:
> 1. Preventing Overfitting: By projecting Q onto a hyperspherical space, L2-norm constrains the representation space.
> 2. Stabilizing Softmax Outputs: L2-normalized Q prevents numerical instability, which is crucial in functions involving exponentiation, such as softmax. Unlike layer-norm, L2-norm directly controls vector magnitudes. Without L2-norm, **large logits can cause sharp distributions and large gradients**, destabilizing training. L2-norm addresses these issues, a capability that layer-norm lacks.
> 3. Enhancing Training Stability: Our experiments show that applying L2-norm results in more stable gradients during training. Specifically, **the grad norm remains below 2 with L2-norm, while it can reach around 70 without it, leading to training instability.**
>
> Our experiments indicated that omitting L2-norm on Q significantly degrades performance while applying it to K has negligible effects.
>
> |l2norm position|cifar100|ImageNet 1k|
> |-|-|-|
> |Only Q|88.4|79.9|
> |Only K|87.5 |-|
> |Q & K|88.3|-|
> |w/o|87.5|79.2|

---

### Author Rebuttal · Authors · 2024-08-07

Dear Reviewers,

We would like to express our gratitude for your insightful feedback and suggestions, which have been instrumental in updating and enhancing our submission. We are particularly grateful for your recognition of the novelty of our approach (R 7BZ9, ysa3), the thoroughness of our experiments (R 7BZ9, qYPk, ysa3), the substantial technical contributions (R qYPk), and the clarity and quality of our writing (R qYPk, ysa3). Your positive comments on these aspects are greatly appreciated.

We kindly invite you to review our author rebuttal. For each point raised by the reviewers, we have provided detailed responses. We hope our responses adequately address your concerns. If you have any further questions, please do not hesitate to bring them to our attention. We sincerely appreciate your valuable feedback.

Best regards,

Authors

---

### Decision · Program_Chairs · 2024-09-25

**Decision:**

Accept (poster)

**Comment:**

In the initial reviews, all reviewers acknowledged the novelty of this paper and the thoroughness of the experiments. However, one reviewer pointed out that essential experiments were missing, such as a comparison to a Soft MoE pre-trained on ImageNet21k and fine-tuned on the evaluation dataset, or a model pre-trained on ImageNet21k, upcycled to an MoE, and fine-tuned (with only the router and experts' MLP layers) on the evaluation dataset. During the rebuttal phase, the authors addressed these concerns with additional experiments, leading to positive feedback from all reviewers. AC has also carefully reviewed the paper, the reviews, and the rebuttal, and concurs with the reviewers' opinions. Therefore, AC recommends acceptance. It would be highly beneficial to include the additional results from the rebuttal phase in the final version.